# Average Sensitivity of Euclidean $k$-Clustering

**Yuichi Yoshida**
National Institute of Informatics
JST, PRESTO
yyoshida@nii.ac.jp

**Shinji Ito**
NEC
i-shinji@nec.com

## Abstract

Given a set of $n$ points in $\mathbb{R}^d$, the goal of Euclidean $(k, \ell)$-clustering is to find $k$ centers that minimize the sum of the $\ell$-th powers of the Euclidean distance of each point to the closest center. In practical situations, the clustering result must be stable against points missing in the input data so that we can make trustworthy and consistent decisions. To address this issue, we consider the average sensitivity of Euclidean $(k, \ell)$-clustering, which measures the stability of the output in total variation distance against deleting a random point from the input data. We first show that a popular algorithm $k$-MEANS++ and its variant called $D^\ell$-SAMPLING have low average sensitivity. Next, we show that any approximation algorithm for Euclidean $(k, \ell)$-clustering can be transformed to an algorithm with low average sensitivity while almost preserving the approximation guarantee. As byproducts of our results, we provide several algorithms for consistent $(k, \ell)$-clustering and dynamic $(k, \ell)$-clustering in the random-order model, where the input points are randomly permuted and given in an online manner. The goal of the consistent setting is to maintain a good solution while minimizing the number of changes to the solution during the process, and that of the dynamic setting is to maintain a good solution while minimizing the (amortized) update time.

## 1 Introduction

*Euclidean $(k, \ell)$-clustering* is a popular problem used to compute clustering of points in the Euclidean space, where $k$ is a positive integer and $\ell \geq 1$. In this problem, given a set of points $X = \{x_1, \ldots, x_n\}$ in $\mathbb{R}^d$, we are to compute a set $Z = \{z_1, \ldots, z_k\}$ of $k$ points in $\mathbb{R}^d$ that minimizes

$$\mathrm{cost}_Z^\ell(X) := \sum_{i=1}^n \mathrm{cost}_Z^\ell(x_i),$$

where $\mathrm{cost}_Z^\ell(x_i) := \min_{j=1,\ldots,k} \|x_i - z_j\|_2^\ell$ is the $\ell$-th power of the Euclidean distance between $x$ and the closest point in $Z$. Let $\mathrm{opt}^\ell(X) = \min_Z \mathrm{cost}_Z^\ell(X)$, where the minimum is over sets of $k$ points in $\mathbb{R}^d$ (not $X$). This problem is called $k$-*means clustering* when $\ell = 2$ [18] and $k$-*medians clustering* when $\ell = 1$. Although providing an exact solution to Euclidean $(k, \ell)$-clustering is NP-hard [3, 7] (even when $\ell = 2$), polynomial-time approximation schemes (PTASes) do exist [13]. In practice, an algorithm called $D^\ell$-SAMPLING is often used, which computes $O(2^\ell \log k)$-approximate solution in polynomial time [4]. When $\ell = 2$, this algorithm is often called $k$-MEANS++.

In practical situations, it is reasonable to assume that the input contains some noise. If the clustering results are altered by the slightest change in the input, the interpretability and reliability of the clustering results are compromised, and hence the clustering result should be stable. Stability is also important when making marketing strategies or product production plans building on the clustering result because changing decisions can be very costly. In this work, we focus on a situation in which some points are missing from the input, and we consider average sensitivity to measure the stability

36th Conference on Neural Information Processing Systems (NeurIPS 2022).

of clustering algorithms, which is defined below. For a set of points $X = \{x_1, \ldots, x_n\}$ and $i \in [n]$, let $X^{(i)}$ denote the set $\{x_1, \ldots, x_{i-1}, x_{i+1}, \ldots, x_n\}$, where $[n] := \{1, 2, \ldots, n\}$. Then, the *average sensitivity* of a (randomized) algorithm $A$ on $X$ (with respect to the total variation distance) [23] is defined as

$$\beta(A, X) := \frac{1}{n} \sum_{i=1}^{n} d_{\text{TV}}(A(X), A(X^{(i)})), \tag{1}$$

where $d_{\text{TV}}(A(X), A(X^{(i)}))$ is the total variation distance between the distributions of $A(X)$ and $A(X^{(i)})$. Average sensitivity has been studied on various problems such as maximum matching [25], spectral clustering [21], dynamic programming [15, 16], and network centralities [20].

It is crucial to take the average in (1) instead of the maximum, which measures the worst-case sensitivity against deleting points. To see this, let us consider the case $k = 2$ and a set of points $X$ consisting of two clusters each having $(n-1)/2$ points and one distant point. Then any good solution must contain the distant point and a middle point between the two clusters. However, for the set of points $X'$ obtained from $X$ by deleting the distant point, any good solution must contain a point from each cluster. Hence, the worst-case sensitivity is $\Omega(1)$. By contrast, the average sensitivity can be $O(1/n)$ because the contribution of deleting the distant point to the average sensitivity is $O(1/n)$.

## 1.1 Our contributions

**Algorithms with low average sensitivity** We first show that $D^{\ell}$-SAMPLING has average sensitivity $O(k/n)$ (Section 2). This matches the lower bound of $\Omega(k/n)$ on the average sensitivity of any algorithm with a finite approximation ratio (Appendix A).

We then show that any approximation algorithm can be transformed to an algorithm with low average sensitivity while almost preserving the approximation ratio. Specifically, given an $\alpha$-approximation algorithm $A$ for $(k, \ell)$-clustering and $\epsilon > 0$, we can transform it into a $(1 + \epsilon)\alpha$-approximation algorithm $A'$ with average sensitivity $\text{poly}(d, k, 2^{\ell}, 1/\epsilon)/n$ (Section 3). For example, if we plug in a PTAS for $(k, \ell)$-clustering [13] as $A$, we obtain a PTAS with average sensitivity $\text{poly}(d, k, 2^{\ell}, 1/\epsilon)/n$ as $A'$. In our transformation, we first construct a coreset [2, 11] of the input, a subset of the input that preserves the objective function, and then apply the approximation algorithm $A$.[1] We notice that we can construct the coreset with low average sensitivity, and hence any approximation algorithm applied on it has low average sensitivity. This approach is new to the design of algorithms with low average sensitivity.

**Consistent clustering** In consistent Euclidean $(k, \ell)$-clustering, $n$ points in $\mathbb{R}^d$ arrive online, and the goal is to maintain a good solution, while minimizing *inconsistency*, which is defined as the total number of points changed in the solution, i.e., $\sum_{t \in [n-1]} |Z_t \triangle Z_{t+1}|$, where $Z_t$ is the output of the algorithm after the $t$-th point arrived.

We consider the random-order model, in which the input points are randomly permuted. We observe that an average sensitivity analysis can be used to obtain consistent algorithms. Specifically, we show that, given an input $X$, an index $i \in [n]$, and the output $Z$ of an algorithm $A$ for Euclidean $(k, \ell)$-clustering for $X$, we can compute an output $Z'$ so that the probability $Z$ and $Z'$ disagree is at most the total variation distance between $A(X)$ and $A(X^{(i)})$ in expectation over $Z$. Using this connection, we show that a variant of $D^{\ell}$-SAMPLING is an $O(2^{\ell} \log k)$-approximation algorithm with inconsistency $O(k^2 \log n)$. Also, we show that the abovementioned PTAS with low average sensitivity can be used to obtain a $(1 + \epsilon)$-approximation algorithm with inconsistency $\text{poly}(d, k, 2^{\ell}, 1/\epsilon) \cdot \log n$.

Lattanzi and Vassilvitskii [17] considered consistent algorithms for *metric $(k, \ell)$-clustering*, for which the underlying metric can be arbitrary but the output points should be selected from the input points. They showed an $2^{O(\ell)}$-approximation algorithm with inconsistency $O(k^2 \log^4 n)$, where $n$ is the number of points. Subsequently, Fichtenberger et al. [8] gave an algorithm for metric $(k, 1)$-clustering with approximation ratio $O(1)$ and inconsistency $O(k \cdot \text{poly}(\log(n\Delta)))$, where $\Delta$ is the aspect ratio of the input, that is, the ratio between the largest and smallest distances. We note that these results are

---

[1] We note that the term *sensitivity* is used as the name of a well studied notion in the study of coresets, which measures the importance of a point in the input (See Section 3.1), and the reader should not confuse it with average sensitivity, which measures the solution change caused by deleting points from the input.

incomparable to ours because in Euclidean $(k, \ell)$-clustering we can output points that do not appear in the input, which may significantly reduces the objective value.

**Dynamic clustering**   In dynamic (or incremental) Euclidean $(k, \ell)$-clustering, points arrive online, and the goal is to maintain a good solution, while minimizing the (amortized) time required to update the solution. For this problem, a few heuristics have been proposed [1], but to the best of our knowledge, there is no known dynamic algorithm with a theoretical guarantee on its update time.

As with consistent clustering, we consider the random-order model. We show that the consistent version of $D^\ell$-SAMPLING can be implemented such that the amortized update time is $O(dk + (k + \log n)k \log n)$. Also, we show that the abovementioned consistent PTAS can be implemented such that the amortized update time is $\mathrm{poly}(d, k, \ell, 1/\epsilon) + 2^{\mathrm{poly}(d, k, 2^\ell, 1/\epsilon)} \log^2 n$.

We note that several dynamic algorithms that maintain coresets for metric $k$-clustering have been proposed [10, 12]. However, it is not clear whether these algorithms can be used to solve dynamic Euclidean $(k, \ell)$-clustering.

## 1.2   Related work

**Differentially private $k$-means/medians clustering**   Differential privacy is a popular notion used to measure privacy risks. Although we do not define it formally here, we note that, if an algorithm is $\beta$-differentially private, then its average sensitivity is at most $\beta$ (see, e.g., [23]). In [9], it is shown that any polynomial-time $\alpha$-approximation algorithm for $k$-means/medians clustering can be converted into an $\beta$-differentially private algorithm that, given a set $X$ of points in a $d$-dimensional unit ball, outputs a solution $Z$ such that

$$\mathrm{cost}_Z^\ell(X) \leq (1 + \epsilon)\alpha \cdot \mathrm{opt}^\ell(X) + \tilde{O}_\alpha \left( \frac{k\sqrt{d} + k^{O_\alpha(1)}}{\epsilon} \right),$$

where $O_\alpha(\cdot)$ means that $\alpha$ is regarded as a constant and $\tilde{O}(\cdot)$ suppresses polylogarithmic dependence on $n$, $d$, $k$, and $\beta$. We note that the additive error is inevitable while obtaining $\beta$-differentially private algorithms for $k$-means/medians clustering [22], whereas we do not need it to bound average sensitivity.

**Stability**   Let $\mathcal{D}$ be a distribution over points in $\mathbb{R}^d$, and suppose that we want to find a cluster structure in $\mathcal{D}$ using an algorithm $A$. In many cases, we do not know the parameter $k$ that best describes $\mathcal{D}$. A popular method for selecting $k$ is based on the following stability arguments. That is, we select the $k$ that minimizes the *instability* of $A$,

$$\mathrm{Instab}(k, n) := \mathop{\mathbf{E}}_{X, X' \sim \mathcal{D}^n} d(A(X, k), A(X', k)),$$

where $A(X, k)$ is the output of $A$ on a set of points $X$ and the number of clusters $k$, and $d(\cdot, \cdot)$ is the distance between two clusterings, which can be defined in various ways.

The main focus of the literature on instability is characterizing $\mathcal{D}$ such that $\mathrm{Instab}(k, n) \to 0$ ($n \to \infty$) holds when $A$ is the algorithm that outputs an optimal solution, because we can then claim that the $k$ captures the cluster structure in $\mathcal{D}$ well. Refer to [24] for a survey on this topic. A notable difference of our work is that we focus on *designing* an algorithm with low average sensitivity for *any* input set $X$, and not characterizing $X$ such that a certain algorithm has low average sensitivity.

## 2   Average sensitivity of $D^\ell$-sampling

In this section, we analyze the average sensitivity of a popular algorithm for Euclidean $(k, \ell)$-clustering called $D^\ell$-sampling, which has approximation ratio $O(2^\ell \log k)$ [4]. This algorithm starts with a uniformly sampled point in the input $X$, and then iteratively adds points to the solution $Z$: at each step, a point $x \in X$ is sampled with probability proportional to $\mathrm{cost}_Z^\ell(x)$ (See Algorithm 1 for details). Throughout, we use the symbol $n$ to denote the number of points in the input set $X$. Our goal is to show the following:

**Theorem 2.1.** *The average sensitivity of $D^\ell$-SAMPLING$(X, k, \ell)$ is $O(k/n)$.*

**Algorithm 1:** $D^\ell$-sampling for Euclidean $(k, \ell)$-clustering

---

**1** **Procedure** $D^\ell$-SAMPLING$(X, k, \ell)$

**2**      Sample $x \in X$ uniformly at random and then set $Z = \{x\}$;

**3**      **for** $i = 2, \ldots, k$ **do**

**4**          Let $\mathcal{D}_{X,Z}$ be the probability distribution over $X$ such that $x \in X$ is sampled with
         probability $p_{X,Z}(x) := \frac{\mathrm{cost}^\ell_Z(x)}{\mathrm{cost}^\ell_Z(X)}$;

**5**          Sample $x \in X$ from $\mathcal{D}_{X,Z}$ and add it to $Z$.

**6**      **return** $Z$.

---

We start by analyzing the average sensitivity of the process of sampling from $\mathcal{D}_{X,Z}$:

**Lemma 2.2.** *The average sensitivity of the process of sampling from $\mathcal{D}_{X,Z}$ is $O(1/n)$.*

*Proof.* For notational simplicity, we use $p(x)$ and $p^{(i)}(x)$ to denote $p_{X,Z}(x)$ and $p_{X^{(i)},Z}(x)$, respectively (Recall that $X^{(i)} = \{x_1, \ldots, x_{i-1}, x_{i+1}, \ldots, x_n\}$). Note that $p(x) = \mathrm{cost}^\ell_Z(x)/\mathrm{cost}^\ell_Z(X)$.

We start by analyzing the change in the sampling probability $p(x)$ caused by deleting a point from $X$. For any $i \in [n]$ and $x \in X^{(i)}$, we have

$$p^{(i)}(x) - p(x) = \frac{\mathrm{cost}^\ell_Z(x)}{\mathrm{cost}^\ell_Z(X^{(i)})} - \frac{\mathrm{cost}^\ell_Z(x)}{\mathrm{cost}^\ell_Z(X)} = \frac{\mathrm{cost}^\ell_Z(x_i) \cdot \mathrm{cost}^\ell_Z(x)}{\mathrm{cost}^\ell_Z(X^{(i)}) \cdot \mathrm{cost}^\ell_Z(X)},$$

where the last equality follows from $\mathrm{cost}^\ell_Z(X) - \mathrm{cost}^\ell_Z(X^{(i)}) = \mathrm{cost}^\ell_Z(x_i)$. Then, we have

$$\sum_{i=1}^n \sum_{x \in X^{(i)}} \left| p(x) - p^{(i)}(x) \right| = \sum_{i=1}^n \sum_{x \in X^{(i)}} \frac{\mathrm{cost}^\ell_Z(x_i) \cdot \mathrm{cost}^\ell_Z(x)}{\mathrm{cost}^\ell_Z(X^{(i)}) \cdot \mathrm{cost}^\ell_Z(X)} = \sum_{i=1}^n \frac{\mathrm{cost}^\ell_Z(x_i)}{\mathrm{cost}^\ell_Z(X)} = 1.$$

Noting that $x_i$ does not exist in $X^{(i)}$, the average sensitivity of the process of sampling from $\mathcal{D}_{X,Z}$ is

$$\frac{1}{n} \sum_{i=1}^n \left( p(x_i) + \sum_{x \in X^{(i)}} \left| p(x) - p^{(i)}(x) \right| \right) = \frac{1}{n} + \frac{1}{n} = O\left(\frac{1}{n}\right). \qquad \square$$

We next analyze the average sensitivity of $D^\ell$-SAMPLING and prove Theorem 2.1.

*Proof of Theorem 2.1.* Let $z_1, \ldots, z_k \in X$ be the sequence of points added to $Z$. Note that they are random variables. We use $z_j^{(i)}$ to denote $z_j$ for the input $X^{(i)}$. First, we have

$$\frac{1}{n} \sum_{i=1}^n d_{\mathrm{TV}}(z_1, z_1^{(i)}) = \frac{1}{n} \sum_{i=1}^n \left( \frac{1}{n} + \sum_{x \in X^{(i)}} \left( \frac{1}{n-1} - \frac{1}{n} \right) \right) = \frac{1}{n} \sum_{i=1}^n \left( \frac{1}{n} + \frac{1}{n} \right) = O\left(\frac{1}{n}\right). \tag{2}$$

For $j \in [k]$ and $q_1, \ldots, q_{k-1} \in X$, we denote by $\{z_j \mid z_1 = q_1, \ldots, z_{j-1} = q_{j-1}\}$ the random variable $z_j$ conditioned on $z_1 = q_1, \ldots, z_{j-1} = q_{j-1}$. We define $z_1^{(i)}, \ldots, z_k^{(i)}$ and $\{z_j^{(i)} \mid z_1^{(i)} = q_1, \ldots, z_{j-1}^{(i)} = q_{j-1}\}$ similarly using $X^{(i)}$.

Then by Lemma 2.2, we have

$$\frac{1}{n} \sum_{i=1}^n d_{\mathrm{TV}}\left( \{z_j \mid z_1 = q_1, \ldots, z_{j-1} = q_{j-1}\}, \{z_j^{(i)} \mid z_1^{(i)} = q_1, \ldots, z_{j-1}^{(i)} = q_{j-1}\} \right) = O\left(\frac{1}{n}\right). \tag{3}$$

for every $j \in \{2, \ldots, k\}$ and $q_1, \ldots, q_{k-1} \in X$.

Given (2) and (3), by induction, we obtain

$$\frac{1}{n} \sum_{i=1}^n d_{\mathrm{TV}}\left( \{z_1, \ldots, z_k\}, \{z_1^{(i)}, \ldots, z_k^{(i)}\} \right) = O\left(\frac{k}{n}\right). \qquad \square$$

---

**Algorithm 2:** Coreset construction by importance sampling

---
**1 Procedure** CORESET($X, k, \ell, s, m$)
**2**     Let $C$ be an empty set;
**3**     **for** $m$ *times* **do**
**4**         Sample $x \in X$ with probability $p(x) := s(x)/\sum_{x' \in X} s(x')$ ;
**5**         $C \leftarrow C \cup \{x\}$;
**6**         Sample $\tilde{p}$ from $[p(x), (1 + \epsilon/2)p(x)]$ uniformly at random ;
**7**         **if** $w(x)$ *is undefined* **then** $w(x) \leftarrow 1/\tilde{p}$. ;
**8**         **else** $w(x) \leftarrow w(x) + 1/\tilde{p}$. ;
**9**     **return** $(C, w)$.

---

## 3 General transformation

In this section, we show that any approximation algorithm for Euclidean $(k, \ell)$-clustering can be converted into an algorithm with low average sensitivity. Specifically, we show the following:

**Theorem 3.1.** *Let $A$ be an $\alpha$-approximation algorithm for Euclidean $(k, \ell)$-clustering with time complexity $T(n, d, k, \ell)$, where $n$ is the number of points and $d$ is the dimension of the Euclidean space. Then for any $\epsilon, \delta > 0$, there exists an algorithm for Euclidean $(k, \ell)$-clustering such that*

- *it outputs $(1 + \epsilon)\alpha$-approximation with probability at least $1 - \delta$,*

- *its average sensitivity is*

$$\widetilde{O}\left(\frac{2^{2\ell}k}{\epsilon^3 n}\left(dk\ell + \log\frac{1}{\delta}\right)\right),$$

 *where $\widetilde{O}(\cdot)$ suppresses a polylogarithmic factor in $k$ and $1/\epsilon$, and*

- *it runs in* $\mathrm{poly}(n, d, k) + T(\mathrm{poly}(d, k, 1/\epsilon, \log(1/\delta)), d, k, \ell)$.

For example, by plugging in a PTAS to Theorem 3.1 [13], we obtain a PTAS with the same average sensitivity as in the theorem.

Our transformation constructs a coreset for the input, which is a small subset that approximately preserves the objective function, and then apply the given algorithm. We first describe known results about coresets in Section 3.1, and then provide an algorithm that constructs coresets with low average sensitivity in Section 3.2. Finally, we prove Theorem 3.1 in Section 3.3.

### 3.1 Preliminaries

A *weighted set* is a pair of a set $X$ and a weight function $w : X \to \mathbb{R}_+$. We say that $(X, w)$ is a *weighted subset* of $Y$ if $X \subseteq Y$. For a weighted set $(X, w)$ and a set of points $Z$, we define $\mathrm{cost}_Z^\ell((X, w))$ as $\sum_{x \in X} w(x)\mathrm{cost}_Z^\ell(x)$. Note that $\mathrm{cost}_Z^\ell((X, w)) = \mathrm{cost}_Z^\ell(X)$ when $w \equiv 1$. A coreset for Euclidean $(k, \ell)$-clustering is defined as follows:

**Definition 3.2** (Coreset for Euclidean $(k, \ell)$-clustering [2, 11]). Let $X$ be a set of points in $\mathbb{R}^d$. For $\epsilon > 0$, we say that a weighted subset $(C, w)$ of $X$ is an $\epsilon$-*coreset* of $X$ if for any set $Z \subseteq X$ of $k$ points, we have

$$(1 - \epsilon)\mathrm{cost}_Z^\ell(X) < \mathrm{cost}_Z^\ell((C, w)) < (1 + \epsilon)\mathrm{cost}_Z^\ell(X).$$

Importance sampling is a popular approach to obtain a small coreset. Given a set of points $X$ in $\mathbb{R}^d$, an importance function $s : X \to \mathbb{R}_+$, and a positive integer $m > 0$, we sample a point $x \in X$ with probability proportional to $s(x)$ and then set its weight to be the reciprocal of the sampling probability, and then we repeat this process $m$ times to construct a coreset. For technical reasons, we slightly perturb the weight of the points. See Algorithm 2 for details.

A popular choice for the importance is the sensitivity with respect to the input. In the context of Euclidean $(k, \ell)$-clustering, for a set $X$ of points in $\mathbb{R}^d$, the *sensitivity* of $x \in X$ is defined to be

$$\sigma_X(x) := \max_{Z \subseteq X : |Z| = k} \frac{\mathrm{cost}_Z^\ell(x)}{\mathrm{cost}_Z^\ell(X)}.$$

Indeed, any upper bound on sensitivity can be used to construct a coreset:

**Theorem 3.3** (Theorem 5.5 in [5], restated. See also [13]). *Let $X$ be a set of points in $\mathbb{R}^d$ and $s : X \to \mathbb{R}_+$ be an arbitrary upper bound on $\sigma_X$. Let $(C, w) = \text{CORESET}(X, k, \ell, s, m)$ with*

$$m \geq \frac{cS}{\epsilon^2} \left( dk \log S + \log \frac{1}{\delta} \right), \tag{4}$$

*where $c > 0$ is some universal constant and $S = \sum_{x \in X} s(x)$. Then, with probability at least $1 - \delta$, $(C, w)$ is an $\epsilon$-coreset of $X$.*

We note that the slight modification to the assigned weights preserves the objective value to within the factor of $1/(1 + \epsilon/2)$, and hence the original analysis of Theorem 3.3 goes through.

The reason that we consider upper bounds on sensitivity in Theorem 3.3 is that no algorithm for exactly computing sensitivity is known. The following lemma shows that any approximate solution can be used to obtain an upper bound.

**Lemma 3.4** ([13]). *Let $X$ be a set of points in $\mathbb{R}^d$ and let $Z \subseteq X$ be an $\alpha$-approximate solution to Euclidean $(k, \ell)$-clustering on $X$. For $x \in X$, let $Z_x = \{y \in X : \exists z \in Z \text{ s.t. } \|x - z\|_2^\ell = \text{cost}_Z^\ell(x), \|y - z\|_2^\ell = \text{cost}_Z^\ell(y)\}$ be the set of all points in the same cluster as $x$. Then, the sensitivity $\sigma_X(x)$ is bounded from above by*

$$s(x) := 2^{2\ell+2} \alpha^2 \cdot \left( \frac{\text{cost}_Z^\ell(x)}{\text{cost}_Z^\ell(X)} + \frac{1}{|Z_x|} \right).$$

*Furthermore, we have $S = \sum_{x \in X} s(x) \leq 2^{2\ell+3} \alpha^2 k$.*

### 3.2 Coreset construction with low average sensitivity

In this section, we consider constructing coresets with low average sensitivity. Specifically, we show the following:

**Lemma 3.5.** *For any $\epsilon, \delta > 0$, there exists a polynomial-time algorithm that, given a set $X$ of points in $\mathbb{R}^d$ and a positive integer $k$ and $\ell \geq 1$, outputs a weighted subset $(C, w)$ such that (i) $(C, w)$ is an $\epsilon$-coreset of $X$ with probability at least $1 - \delta$, and (ii) we have*

$$|C| \leq m := \widetilde{O} \left( \frac{2^{2\ell} k}{\epsilon^2} \left( dk\ell + \log \frac{1}{\delta} \right) \right).$$

*The average sensitivity of the algorithm is at most $O(\frac{m}{\epsilon n})$.*

Our algorithm is as follows. We first compute a set $Z \subseteq X$ by calling $D^\ell$-SAMPLING$(X, k, \ell)$. Then, we compute the upper bounds on the sensitivities of points in $X$ as in Lemma 3.4, which we denote by $s_{X,Z}(x)$ for $x \in X$. Finally, we apply Theorem 3.3 using $s_{X,Z}(x)$ with the smallest $m$ satisfying the condition in the statement, denoted $m_Z$. Note that $s_{X,Z}(x)$ and $m_Z$ are random variables because the output $Z$ of $D^\ell$-SAMPLING is a random variable. It is clear that this algorithm outputs an $O(\epsilon)$-coreset with probability at least $1 - \delta$. The proof of Lemma 3.5 is deferred to Appendix B.

### 3.3 Proof of Theorem 3.1

Our algorithm is as follows: We first compute an $\epsilon$-coreset $(C, w)$ of the input $X$ using Lemma 3.5, and then apply the $\alpha$-approximation algorithm $A$ given in the statement to the coreset. The analysis of approximation ratio and average sensitivity is trivial.

We discuss the time complexity of the algorithm above. Suppose that the algorithm $A$ runs in $T(n, d, k, \ell)$ time. Then, we can compute the coreset $(C, w)$ in $\text{poly}(n, d, k)$ time. To run the algorithm $A$, we need $T(|C|, d, k, \ell) = T(\text{poly}(d, k, 1/\epsilon, \log(1/\delta)), d, k, \ell)$ time. Hence, in total, we need $\text{poly}(n, d, k) + T(\text{poly}(d, k, 1/\epsilon, \log(1/\delta)), d, k, \ell)$ time.

# 4 Consistent clustering in the random-order model

In this section, we show that algorithms for Euclidean $(k, \ell)$-clustering with low average sensitivity can be used to design algorithms for consistent Euclidean $(k, \ell)$-clustering in the random-order model. In Section 4.1, we show that an algorithm with low average sensitivity satisfying a certain condition can be transformed into a consistent algorithm. Then, we see in Section 4.2 and Appendix C that the algorithms discussed in Sections 2 and 3, respectively, satisfy the condition and can be transformed into consistent algorithms.

## 4.1 Low average sensitivity to consistency

We start with the following definition.

**Definition 4.1.** Let $X = \{x_1, \ldots, x_n\}$ be a set of points in $\mathbb{R}^d$. For $i \in [n]$, let $\mathcal{Z}$ and $\mathcal{Z}^{(i)}$ be two distributions over subsets of $X$ and $X^{(i)}$, respectively, of size $k$. Suppose that $\sum_{i=1}^n d_{\mathrm{TV}}(\mathcal{Z}^{(i)}, \mathcal{Z})/n \leq \beta$ holds. Then, we say that a probability transportation for $\mathcal{Z}$ of average sensitivity $\beta$ is *computable* if the following holds:

1. For each $i \in [n]$, there is a distribution $\mathcal{D}^{(i)}$ over pairs of sets such that its marginal distributions on the first and second coordinates are equal to $\mathcal{Z}^{(i)}$ and $\mathcal{Z}$, respectively.

2. $\sum_{i=1}^n \Pr_{(Z^{(i)}, Z) \sim \mathcal{D}^{(i)}}[Z^{(i)} \neq Z]/n \leq \beta$.

3. For each $i \in [n]$, there is a (randomized) algorithm that takes a set $Z^{(i)}$ and returns a set $Z$, such that if the input $Z^{(i)}$ is distributed as the distribution $\mathcal{Z}^{(i)}$, then the distribution of the pair $(Z^{(i)}, Z)$ is equal to $\mathcal{D}^{(i)}$. In particular, $Z$ is distributed as $\mathcal{Z}$.

The following gives a general transformation from algorithms with low average sensitivity to consistent algorithms where a probability transportation is computable:

**Lemma 4.2.** *Let $A$ be a (randomized) algorithm for Euclidean $(k, \ell)$-clustering with approximation ratio $\alpha$ and average sensitivity at most $\beta(n)$ for inputs of size $n$. Suppose that the probability transportation for $A(X)$ of average sensitivity $\beta(n)$ is computable for any set $X$ of $n$ points. Then, there is an algorithm for consistent Euclidean $(k, \ell)$-clustering in the random-order model such that*

- *the expected approximation ratio is $\alpha$ for each step, where the expectation is taken over the randomness of the algorithm, and*

- *the expected inconsistency is at most $k \cdot \sum_{i=1}^n \beta(i)$, where the expectation is taken over both the randomness of the algorithm and the arrival order of the input.*

*Proof.* Let $X = \{x_1, \ldots, x_n\}$ be a set of points in $\mathbb{R}^d$, and let $\sigma$ be a random permutation of $[n]$. The points are given in the order $x_{\sigma(1)}, \ldots, x_{\sigma(n)}$. We first compute $Z_{\sigma,1}$ by running $A$ on $x_{\sigma(1)}$. For $t \geq 2$, we compute $Z_{\sigma,t}$ from $Z_{\sigma,t-1}$ using the probability transportation for $A(\{x_{\sigma(1)}, \ldots, x_{\sigma(t)}\})$ given in the statement. Let $\mathcal{D}_{\sigma,t}$ be the joint distribution given by the probability transportation such that the marginal on the first and second coordinates are $Z_{\sigma,t}$ and $Z_{\sigma,t-1}$, respectively. Our output sequence is $Z_{\sigma,1}, \ldots, Z_{\sigma,n}$.

We observe that the expected approximation ratio is $\alpha$ because the distribution of $Z_{\sigma,t}$ is the same as the output distribution of $A(\{x_{\sigma(1)}, \ldots, x_{\sigma(t)}\})$. The expected number of times that the output changes is bounded as

$$\mathbf{E}_\sigma \left[ \sum_{t=1}^n \Pr[Z_{\sigma,t-1} \neq Z_{\sigma,t}] \right] = \mathbf{E}_\sigma \left[ \sum_{t=1}^n \Pr_{(Z,Z') \sim \mathcal{D}_{\sigma,t}}[Z \neq Z'] \right]$$

$$= \sum_{t=1}^n \mathbf{E}_\sigma \left[ \Pr_{(Z,Z') \sim \mathcal{D}_{\sigma,t}}[Z \neq Z'] \right] = \sum_{t=1}^n \frac{1}{n} \sum_{i=1}^n \mathbf{E}_\sigma \left[ \Pr_{(Z,Z') \sim \mathcal{D}_{\sigma,t}}[Z \neq Z'] \mid \sigma_t = i \right] \leq \sum_{t=1}^n \beta(t).$$

As we have $\mathbf{E} |Z_{\sigma,t-1} \triangle Z_{\sigma,t}| \leq k \cdot \Pr[Z_{\sigma,t-1} \neq Z_{\sigma,t}]$, the claim holds. $\square$

## 4.2 Consistent $D^\ell$-sampling

In this section, we propose a consistent algorithm that is a slight modification of $D^\ell$-SAMPLING, and show the following:

**Theorem 4.3.** *There exists a polynomial-time $O(\log k)$-approximation algorithm for consistent Euclidean $(k, \ell)$-clustering in the random-order model with inconsistency $O(k^2 \log n)$.*

For a probability distribution $\mathcal{Z}$, we say that we have *sampling access* to $\mathcal{Z}$ if we can sample a value from $\mathcal{Z}$. For $z$ in the support of $\mathcal{Z}$, let $\mathcal{Z}(z)$ denote the probability that $z$ is sampled from $\mathcal{Z}$. Then, we say that we have *query access* to $\mathcal{Z}$ if we can obtain the value of $\mathcal{Z}(z)$ for a specified $z$. We use the following lemma.

**Lemma 4.4** ([14]). *Let $\mathcal{Z}, \mathcal{Z}'$ be probability distributions, and suppose that we have sample and query accesses to $\mathcal{Z}$ and $\mathcal{Z}'$. Then, there is an algorithm called LAZYSAMPLING that, given $z$ sampled from $\mathcal{Z}$, outputs $z'$ such that*

- *the distribution of $z'$ is equal to $\mathcal{Z}'$,*

- $\Pr_{z,z'}[z \neq z'] \leq d_{\mathrm{TV}}(\mathcal{Z}, \mathcal{Z}')$,

- *the expected number of samples drawn from $\mathcal{Z}$ and $\mathcal{Z}'$ is $O(1)$.*

- *the expected number of queries to $\mathcal{Z}$ and $\mathcal{Z}'$ is $O(1)$.*

The following lemma provides the probability transportation for $D^\ell$-SAMPLING.

**Lemma 4.5.** *The probability transportation for $D^\ell$-SAMPLING$(X, k, \ell)$ with average sensitivity $O(k/n)$ is computable.*

*Proof.* Fix $i \in [n]$. We provide an algorithm that takes a set $Z^{(i)} = \{z_1^{(i)}, \ldots, z_k^{(i)}\}$ of $k$ points distributed as $D^\ell$-SAMPLING$(X^{(i)}, k, \ell)$ and outputs a set $Z = \{z_1, \ldots, z_k\}$ of $k$ points such that $Z$ is distributed as $D^\ell$-SAMPLING$(X, k, \ell)$ and the probability that $Z^{(i)} \neq Z$ is at most $d_{\mathrm{TV}}(D^\ell$-SAMPLING$(X^{(i)}, k, \ell), D^\ell$-SAMPLING$(X, k, \ell))$.

We first set $z_1 = $ LAZYSAMPLING$(z_1^{(i)}, \mathcal{U}_{X^{(i)}}, \mathcal{U}_X)$, where $\mathcal{U}_{X^{(i)}}$ and $\mathcal{U}_X$ are the uniform distributions over $X^{(i-1)}$ and $X$, respectively. If $z_1 \neq z_1^{(i)}$, then we compute $z_2, \ldots, z_k$ by following the process of $D^\ell$-SAMPLING$(X, k, \ell)$ conditioned on having chosen $z_1$ as the first point. If $z_1 = z_1^{(i)}$, then we compute $z_2 = $ LAZYSAMPLING$(z_2^{(i)}, \mathcal{D}_{X^{(i)}, \{z_1\}}, \mathcal{D}_{X, \{z_1\}})$ (see Algorithm 1 for the definition of $D^\ell$-sampling distribution $\mathcal{D}_{X,Z}$). We repeat this process until we compute $z_k$.

We can observe that the distribution of $Z$ is equal to that of $A(X)$ and that the total variation distance between $Z^{(i)}$ and $Z$ is at most that between $D^\ell$-SAMPLING$(X^{(i)}, k, \ell)$ and $D^\ell$-SAMPLING$(X, k, \ell)$. Hence, the claim holds. □

*Proof of Theorem 4.3.* Consider an algorithm obtained by combining Lemmas 4.2 and 4.5. It is clearly an $O(\log k)$-approximation algorithm, and inconsistency is $k \cdot \sum_{t=1}^n O(k/t) = O(k^2 \log n)$. □

## 5 Dynamic clustering in the random-order model

In this section, we consider algorithms for dynamic Euclidean $(k, \ell)$-clustering in the random-order model. Our algorithms are obtained by carefully implementing the consistent algorithms discussed in Section 4 so that the amortized update time is small. We provide efficient implementations for algorithms discussed in Section 4.2 in Section 5.1. We also show that the consistent transformation discussed in C can be efficiently implemented in D.

## 5.1 Dynamic $D^\ell$-sampling

In this section, we show the following:

**Theorem 5.1.** *There exists an $O(2^\ell \log k)$-approximation algorithm for dynamic Euclidean $(k, \ell)$-clustering in the random-order model with amortized update time $O(dk + (k + \log n)k \log n)$.*

*Proof.* Let $x_1, \ldots, x_n \in \mathbb{R}^d$ be the input points (that arrive in this order). Throughout the process, we maintain points $z_1, \ldots, z_k$ and sequences $S_1, \ldots, S_k$ of real values. When $x_t$ $(t \in [n])$ arrive, for each $j \in [k]$, we update $z_j$ so that its distribution is equal to that of the $j$-th point selected in $D^\ell$-SAMPLING$(X_t, k, \ell)$, where $X_t = \{x_1, \ldots, x_t\}$. Also, we update $S_j$ to be the sequence $\mathrm{cost}^\ell_{Z_{j-1}}(x_1), \ldots, \mathrm{cost}^\ell_{Z_{j-1}}(x_i)$, where $Z_j = \{z_1, \ldots, z_j\}$. When $j = 1$, we regard that $\mathrm{cost}^\ell_{Z_{j-1}}(x_i) = 1$ for every $i \in [t]$. For each $j \in [k]$, we maintain a binary tree on $S_j$ so that we can append a value and compute the sum of values in an arbitrary (consecutive) subsequence of $S_j$ in $O(\log n)$ time. We can support those queries using, say, a segment tree. Note that by querying consecutive sums on $S_j$, we can compute $p_{X_t, Z_{j-1}}(x)$ (see Algorithm 1 for the definition), which is required to simulate $D^\ell$-SAMPLING in $O(\log n)$ time. Also, we can draw a sample from the distribution $\mathcal{D}_{X_t, Z_{j-1}}$ in $O(\log^2 n)$ time using the consecutive sum queries (e.g., draw a number uniformly from $[0, \mathrm{cost}^\ell_{Z_{j-1}}(X_t)]$ and then perform binary search to find the position that the number belongs to). Note that for $j = 1$, the distribution $\mathcal{Z}_{t,1}$ is just a uniform distribution over $X_t$.

We initialize $z_1, \ldots, z_k$ and $S_1, \ldots, S_k$ using the first point $x_1$. That is, $z_1 = \cdots = z_k = x_1$, which is the $k$ points selected by $D^\ell$-SAMPLING$(\{x_1\}, k, \ell)$. Then for each $j \in [k]$, we initialize $S_j$ to be the sequence consisting of a single value $\mathrm{cost}^\ell_{Z_{j-1}}(x_1)/\mathrm{cost}^\ell_{Z_{j-1}}(X_1) = 1$.

Below, we explain how we update $z_1, \ldots, z_k$ and $S_1, \ldots, S_k$ when $x_i$ for $t > 1$ arrives. When updating $z_j$ using LAZYSAMPLING as in Section 4.2, we need sampling and query accesses to $\mathcal{Z}_{t-1,j}$ and $\mathcal{Z}_{t,j}$. We can provide sampling and query accesses to the former by using the current binary tree on $S_j$, and those to the latter by using the binary tree obtained by appending $x_t$ (We can access both by keeping the old one at this step). If $z_j$ is updated to a different point, we rebuild binary trees on $S_j, \ldots, S_k$ from scratch.

Now we analyze the amortized update time. At each step, for each $j \in [k]$, we need $O(dk)$ time to compute $\mathrm{cost}^\ell_{Z_1}(x_t), \ldots, \mathrm{cost}^\ell_{Z_k}(x_t)$, and $O(k \log t)$ time to append $x_t$ to the binary trees on $S_1, \ldots, S_k$. Then, we need $O(k \log^2 n)$ time to update $z_1, \ldots, z_k$. When $z_j$ is replaced, we rebuild binary trees on $S_j, \ldots, S_k$, which takes $O(kt \log t)$ time. Recalling that the probability of replacing some of $z_j$'s at $t$-th step is $O(k/t)$, we need

$$\sum_{t=1}^n (O(dk) + O(k \log t)) + O(k \log^2 n) \cdot n + \sum_{t=1}^n O(kt \log t) \cdot O\left(\frac{k}{t}\right) = O(dkn + (k + \log n)kn \log n)$$

time in total. Hence the amortized update time is $O(dk + (k + \log n)k \log n)$. $\qquad\square$

## 6   Conclusion

We have shown that the average sensitivity of $D^\ell$-SAMPLING is small and that any approximation algorithm can be transformed to one with low average sensitivity. Then, we show that these algorithms can be used to obtain consistent and dynamic algorithms in the random-order model. We believe consistent and dynamic algorithms for other problems can also be obtained via average sensitivity analysis.

It is an interesting open question whether we can remove the dependency on $d$ from the sensitivity bound in Theorem 3.1. A natural idea is to apply dimension reduction, e.g., [19]. However, it is not clear whether we can recover a solution for the original space from that for the reduced space with a small average sensitivity. Another natural idea is to use coresets of size independent of $d$, e.g., [6, 13]. A challenge here is that their constructions are more complicated than the importance sampling used in this work, and their average sensitivity might be high.

## Acknowledgments

Y.Y. is Supported by JST, PRESTO Grant Number JPMJPR192B.

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
