# A Lower bounds

In this section, we show the following lower bound:

**Theorem A.1.** *Any algorithm for Euclidean $(k, \ell)$-clustering with a finite approximation ratio has average sensitivity $\Omega(k/n)$.*

We note that, for algorithms that select $k$ centroids only from the input $X$ (and not from $\mathbb{R}^d \setminus X$), there is a trivial lower bound of $\Omega(k/n)$ because when one of the centroids is deleted, which happens with probability $\Omega(k/n)$, the algorithm must change its output. Theorem A.1 shows that the same lower bound applies even for algorithms that may select centroids from $\mathbb{R}^d \setminus X$.

*Proof of Theorem A.1.* Let $A$ be an algorithm with a finite approximation ratio. Let $X = \{x_1, \ldots, x_n\}$ be a set of points in $\mathbb{R}^n$ such that $x_1, \ldots, x_{k+1}$ are all distinct and $x_{k+1} = x_{k+2} = \cdots = x_n$. Then for any $X^{(i)}$ with $1 \leq i \leq k$, the set $Z_i := \{x_1, \ldots, x_{i-1}, x_{i+1}, \ldots, x_{k+1}\}$ is the unique optimal solution, which gives the objective value zero. Hence to have a finite approximation ratio, the algorithm $A$ must output $Z_i$ on $X^{(i)}$. Let $p_i$ be the probability that the algorithm $A$ outputs $Z_i$ on $X$. Then, the average sensitivity of $A$ on $X$ is

$$\frac{1}{n}\sum_{i=1}^{n} d_{\mathrm{TV}}(A(X), A(X^{(i)})) \geq \frac{1}{n}\sum_{i=1}^{k} d_{\mathrm{TV}}(A(X), A(X^{(i)})) \geq \frac{1}{n}\sum_{i=1}^{k}(1 - p_i)$$

$$\geq \frac{1}{n}(k-1) = \Omega\left(\frac{k}{n}\right). \qquad \square$$

# B Proof of Lemma 3.5

The following useful lemma is implicit in the proof of Lemma 2.3 of [15].

**Lemma B.1.** *For $\epsilon, B, B' > 0$, let $X$ and $X'$ be sampled from the uniform distributions over $[B, (1+\epsilon)B]$ and $[B', (1+\epsilon)B']$, respectively. Then, we have*

$$d_{\mathrm{TV}}(X, X') \leq \frac{1+\epsilon}{\epsilon}\left|1 - \frac{B'}{B}\right|.$$

*Proof of Lemma 3.5.* We now analyze the size of the coreset. As we mentioned, the approximation ratio of $D^\ell$-SAMPLING is $O(2^\ell \log k)$. Also, we have $\mathbf{E}\sum_{x \in X} s_{X,Z}(x) \leq 2^{2\ell+3}O(\log^2 k)k = O(2^{2\ell}k\log^2 k)$ by Lemma 3.4. Hence by the choice of $m_Z$, the size of $C$ is at most

$$O\left(\frac{2^{2\ell}k\log^2 k}{\epsilon^2}\left(dk(\log(2^{2\ell}k\log^2 k)) + \log\frac{1}{\delta}\right)\right) = \tilde{O}\left(\frac{2^{2\ell}k}{\epsilon^2}\left(dk\ell + \log\frac{1}{\delta}\right)\right) \qquad (5)$$

Next, we analyze the average sensitivity. Let $X = \{x_1, \ldots, x_n\}$. Let $Z$ and $Z^{(i)}$ be the outputs of $D^\ell$-SAMPLING on $X$ and $X^{(i)}$, respectively. Then by Theorem 2.1, we have $(1/n)\sum_{i=1}^{n} d_{\mathrm{TV}}(Z, Z^{(i)}) = O(k/n)$. Let $(C, w)$ and $(C^{(i)}, w^{(i)})$ be the coresets constructed for $X$ and $X^{(i)}$, respectively. We have

$$\frac{1}{n}\sum_{i=1}^{n} d_{\mathrm{TV}}((C, w), (C^{(i)}, w^{(i)}))$$

$$= \frac{1}{n}\sum_{i=1}^{n} d_{\mathrm{TV}}(Z, Z^{(i)}) + \frac{1}{n}\sum_{i=1}^{n}\int d_{\mathrm{TV}}(\{(C, w) \mid Z = \tilde{Z}\}, \{(C^{(i)}, w^{(i)}) \mid Z^{(i)} = \tilde{Z}\})\mathrm{d}\tilde{Z}$$

$$= O\left(\frac{k}{n}\right) + \frac{1}{n}\sum_{i=1}^{n}\int d_{\mathrm{TV}}(\{C \mid Z = \tilde{Z}\}, \{C^{(i)} \mid Z = \tilde{Z}\})\mathrm{d}\tilde{Z}$$

$$+ \frac{1}{n}\int\int\sum_{i=1}^{n} d_{\mathrm{TV}}(\{w \mid C = \tilde{C}, Z = \tilde{Z}\}, \{w^{(i)} \mid C^{(i)} = \tilde{C}, Z^{(i)} = \tilde{Z}\})\mathrm{d}\tilde{C}\mathrm{d}\tilde{Z}. \qquad (6)$$

Now, we bound the second term. Let $p(x)$ and $p^{(i)}(x)$ denote the probability of sampling $x$ from $X$ and $X^{(i)}$, respectively, in (one iteration of) CORESET. Conditioned on that $Z = Z^{(i)} = \tilde{Z}$, we have

$$
\sum_{i=1}^{n} \sum_{x \in X^{(i)}} |p(x) - p^{(i)}(x)| = \sum_{i=1}^{n} \sum_{x \in X^{(i)}} \left| \frac{s_{X,\tilde{Z}}(x)}{S_{X,\tilde{Z}}} - \frac{s_{X^{(i)},\tilde{Z}}(x)}{S_{X^{(i)},\tilde{Z}}} \right|
$$

$$
= \sum_{i=1}^{n} \sum_{x \in X^{(i)}} \frac{s_{X,\tilde{Z}}(x)(S_{X,\tilde{Z}} - S_{X^{(i)},\tilde{Z}})}{S_{X,\tilde{Z}} S_{X^{(i)},\tilde{Z}}} = \sum_{i=1}^{n} \sum_{x \in X^{(i)}} \frac{s_{X,\tilde{Z}}(x) \cdot s_{X,\tilde{Z}}(x_i)}{S_{X,\tilde{Z}} S_{X^{(i)},\tilde{Z}}} = \sum_{i=1}^{n} \frac{s_{X,\tilde{Z}}(x_i)}{S_{X,\tilde{Z}}} = 1.
$$

$$(7)$$

Then, we have

$$
\frac{1}{n} \sum_{i=1}^{n} d_{\mathrm{TV}}(\{C \mid Z = \tilde{Z}\}, \{C^{(i)} \mid Z = \tilde{Z}\}) = \frac{m_{\tilde{Z}}}{n} \sum_{i=1}^{n} \left( p(x_i) + \sum_{x \in X^{(i)}} |p(x) - p^{(i)}(x)| \right) = O\left(\frac{m_{\tilde{Z}}}{n}\right).
$$

Hence, the second term of (6) is $O(\mathbf{E}\, m_Z / n)$.

Now we bound the third term of (6). By Lemma B.1, it can be bounded by

$$
\frac{\mathbf{E}\, m_Z}{n} \sum_{i=1}^{n} \left( \sum_{x \in X^{(i)}} \min\left\{p(x), p^{(i)}(x)\right\} \cdot \frac{1+\epsilon}{\epsilon} \left| 1 - \frac{p^{(i)}(x)}{p(x)} \right| \right)
$$

$$
\leq \frac{\mathbf{E}\, m_Z}{n} \sum_{i=1}^{n} \left( \sum_{x \in X^{(i)}} \frac{1+\epsilon}{\epsilon} \left| p(x) - p^{(i)}(x) \right| \right) = O\left( \frac{\mathbf{E}\, m_Z}{\epsilon n} \right),
$$

where the last equality is by (7). By combining above, the average sensitivity of the algorithm is given as

$$
O\left(\frac{k}{n}\right) + O\left(\frac{\mathbf{E}\, m_Z}{n}\right) + O\left(\frac{\mathbf{E}\, m_Z}{\epsilon n}\right) = O\left(\frac{m}{\epsilon n}\right).
$$

By combining the above and (5), the claim follows. $\square$

## C Consistent transformation

In this section, we show that the general transformation discussed in Section 3 can be used to design consistent algorithms in the random-order model. To this end, we first prove the following.

**Lemma C.1.** *Let A be the algorithm of Lemma 3.5. Then, the probability transportation for A with average sensitivity as in Lemma 3.5 is computable.*

*Proof.* Let us fix a set $X$ of $n$ points in $\mathbb{R}^d$ and $i \in [n]$. Then, given a coreset $(C^{(i)}, w^{(i)})$ for $X^{(i)}$, we need to compute a coreset $(C, w)$ for $X$. We apply the probability transportation used in the proof of Theorem 4.3 to compute a set $Z$ of $k$ points for $X$ from a set $Z^{(i)}$ of $k$ points for $X^{(i)}$. If $Z \neq Z^{(i)}$, then we compute the coreset $(C, w)$ by running CORESET. If $Z = Z^{(i)}$, then we recompute points (and weights) added to $C$ by applying LAZYSAMPLING on each point in $C^{(i)}$. This provides a probability transportation, and we can observe that all the conditions of Definition 4.1 are satisfied. $\square$

**Theorem C.2.** *Let A be an $\alpha$-approximation algorithm for Euclidean $(k, \ell)$-clustering. Then for any $\epsilon, \delta > 0$, there exists an algorithm for consistent Euclidean $(k, \ell)$-clustering in the random-order model such that (i) it outputs $(1 + \epsilon)\alpha$-approximation with probability at least $1 - \delta$ at each step, and (ii) its inconsistency is*

$$
\widetilde{O}\left( \frac{2^{2\ell} k^2 \log n}{\epsilon^3} \left( dk\ell + \log \frac{1}{\delta} \right) \right).
$$

*Proof.* We combine Lemma 4.2 and Lemma C.1. The approximation guarantee is clearly satisfied. The inconsistency of the algorithm is $k \cdot \sum_{t=1}^{n} O(\mathbf{E}\, |C|/\epsilon t) = k \log n \cdot O(\mathbf{E}\, |C|/\epsilon)$, and hence the claim holds. $\square$

# D  Dynamic transformation

We show that the consistent transformation discussed in Section C can be implemented in such a way that the amortized update time in the random-order model is small. Specifically, we show the following:

**Theorem D.1.** *Let $A$ be an $\alpha$-approximation algorithm for Euclidean $(k, \ell)$-clustering with time complexity $T(n, d, k, \ell)$. Then for any $\epsilon, \delta > 0$, there exists an algorithm for dynamic Euclidean $(k, \ell)$-clustering in the random-order model that (i) outputs $(1 + \epsilon)\alpha$-approximation with probability at least $1 - \delta$, and (ii) its amortized update time is*

$$O\left( dk + \left( k(k + \log n) + \frac{mT(m, d, k, \ell)}{\epsilon} \right) \log n \right),$$

*where $m = \widetilde{O}\left( \frac{2^{2\ell}k}{\epsilon^2} \left( dk\ell + \log \frac{1}{\delta} \right) \right)$.*

*Proof.* The consistent transformation has two components, that is, $D^\ell$-SAMPLING and coreset construction.

We use the dynamic algorithm of Theorem 5.1 to run the $D^\ell$-SAMPLING part and hence the amortized update time of this part is $O(dk + (k + \log n)k \log n)$.

For the coreset construction part, we maintain a coreset $(C, w)$ and a sequence $S$ storing $s(x_1), \ldots, s(x_t)$, where $s(x)$ is the upper bound on the sensitivity of $x$ as in the proof of Lemma 3.5. We maintain a binary tree on $S$ as with dynamic version of $D^\ell$-SAMPLING. When the output of $D^\ell$-SAMPLING changes after $x_t$ arrives, we recompute $(C, w)$ and the sequence $S$ from scratch. When the output of $D^\ell$-SAMPLING does not change, we append $s(x_t)$ to $S$, and then update the coreset $(C, w)$ using LAZYSAMPLING.

Now we analyze the amortized update time of the coreset construction part. At each step we need $O(|C| \log n)$ time to update $(C, w)$. Also, when the output of $D^\ell$-sampling changes, we need additional $O(t \log t)$ time to reconstruct a binary tree over $S$. Finally, when $(C, w)$ is updated, we need to recompute an optimal solution for $C$, which takes $T(|C|, d, k, \ell)$ time. Recalling that $|C| \leq m$ by Lemma 3.5, in expectation, the total computational time is bounded as

$$\mathbf{E}\left[ O(|C| \log n) \cdot n + \sum_{t=1}^{n} O\left( \frac{k}{t} \right) O(t \log t) + \sum_{t=1}^{n} O\left( \frac{|C|}{\epsilon t} \right) \cdot T(|C|, d, k, \ell) \right]$$

$$= O\left( \left( m + k + \frac{mT(m, d, k, \ell)}{\epsilon} \right) \cdot n \log n \right)$$

$$= O\left( \left( k + \frac{mT(m, d, k, \ell)}{\epsilon} \right) n \log n \right).$$

Combined with the amortized time of dynamic $D^\ell$-SAMPLING, the claim holds. $\qquad\square$