# OpenReview forum: "Average Sensitivity of Euclidean k-Clustering"
_NeurIPS.cc/2022/Conference — NeurIPS 2022 Accept_

### Official Review · Reviewer_k3kz · 2022-07-07

**Rating:** 6
**Confidence:** 4
**Soundness:** 4 excellent
**Presentation:** 3 good
**Contribution:** 4 excellent

**Summary:**

The paper addresses the problems of consistent and dynamic clustering. The metric used for measuring the consistency / stability of an algorithm is the average sensitivity.

The authors first provide a proof that two known clustering and sampling algorithms have very small average sensitivity.
The authors then show how to transform an approximation algorithm for the Euclidean clustering problem into a variation with guaranteed low average sensitivity. This is by providing a novel coreset construction algorithm with low average sensitivity, and then running the original approximation algorithm on the (stable) coreset.

Lastly, based on the low average sensitivity algorithm above, the authors propose algorithms for the consistent and dynamic clustering tasks in the random order models.


**Questions:**

See questions above.

**Limitations:**

Limitations are not clearly mentioned.

**Strengths And Weaknesses:**

Strengths:
- The paper addresses important problem that are very relevant for the inevitability noisy data in the big data era.
- The paper is very well written. The formal claims are concise and are well explained beforehand. The proofs are clear and seem correct.
- The results are interesting; the proposed low average sensitivity algorithm is non-trivial, and makes an interesting use of the known compression scheme known as a coreset.
- One of the conclusions of this work, in my opinion, is important for future coreset construction algorithms: a low average sensitivity coreset construction algorithm yields a low average sensitivity (approximation) algorithm for tackling the problem. This acts as another application for coresets which is not usually discussed in the coresets literature.
- The link to consistent and dynamic algorithms in the random-order model is important.

Weaknesses:
- The "Dynamic clustering" paragraph at Line 76 is, in the best case,  misleading. Many existing coresets for the k-means problem do support fast update on new data arrival. Hence, computing and maintaining such a coreset for the dynamic data also yields a good and fast solution for the problem under the same dynamic-data model.
- The paper lacks a comparison or discussion on other dimensionality reduction techniques, e.g., https://epubs.siam.org/doi/abs/10.1137/18M1209854 .
- No empirical evaluation or comparison to competing methods. The empirical evaluation of the stability of the proposed algorithms and their impact when handling noisy datasets is crucial. Furthermore, the comparison of those variants to the known non-stable algorithms can reveal whether the new variants are indeed neccessary.
- Presentation: The disclaimer on the difference between the two very different "sensitivity" terms (at the bottom of Page 6) should have been made clearly emphasized earlier on. The two terms are very confusing, specially for the readers familiar with the coresets field.
- Limitations and future work are not mentioned.

Minor comment:
- Line 175-176: Can you provide a citation or a more formal claim?

---

> ### Author Response · Authors · 2022-08-02
> **Re: Official Review of Paper11263 by Reviewer k3kz**
>
> Thank you very much for the careful reading and for providing insightful comments.
>
>
> - [Dynamic clustering] Thanks for pointing out the missing comparison.  We checked [1] and references therein. As far as we understand, the dynamic algorithms provided in previous works maintain a coreset of size $O(\mathrm{polylog} n)$ (ignoring dependencies on $\epsilon,k,\ell$). Hence if we apply a PTAS on the coreset to get a $(1+\epsilon)$-approximate solution, the update time becomes $\mathrm{poly}(n)$. By contrast, our update time is $O(\log^2 n)$. Some other dynamic algorithms are known [2, 3], but their bounds depend on the aspect ratio (the ratio of the maximum distance to the minimum distance between pairs of points in the input), which can be huge in general. Also, another notable difference is that the previous works study metric $k$-means, and hence output points should be selected from the input points, whereas we study “Euclidean” $k$-means, and hence the output points can be arbitrarily chosen from $\mathbb{R}^d$.
> - [Dimension reduction] The reason that we used the current dimension reduction is that it can handle the case that the cost is measured by $\|\cdot\|_2^\ell$ for $\ell \geq 1$. The dimension reduction you mentioned is tailored to $\|\cdot \|_2^2$ ($\ell=2$).
> - [Presentation] Thanks for pointing this out. We will revise the manuscript accordingly.
> - [Limitation] One limitation of our online/dynamic algorithms is that they work only in the random-order model. A natural open problems is to improve the average sensitivity bound in Theorem 3.1.
> - [NP-hardness] It is easy to show that computing sensitivity is NP-hard when maximizing each of the cost functions is NP-hard (say, maximizing a polynomial in a bounded region). But because we focus on $k$-clustering in this work, we will instead say “no algorithm for exactly computing sensitivity is known.”
>
> We will revise the manuscript and incorporate the discussion above.
>
> [1] Henzinger, Monika, and Sagar Kale. "Fully-Dynamic Coresets." 28th Annual European Symposium on Algorithms (ESA 2020). Schloss Dagstuhl-Leibniz-Zentrum für Informatik, 2020.
> [2] Bateni, MohammadHossein, et al. "Optimal Fully Dynamic $ k $-Centers Clustering." arXiv preprint arXiv:2112.07050 (2021).
> [3] Goranci, Gramoz, et al. "Fully Dynamic k-Center Clustering in Low Dimensional Metrics." 2021 Proceedings of the Workshop on Algorithm Engineering and Experiments (ALENEX). 2021.

---

> > ### Comment · Reviewer_k3kz · 2022-08-06
> > **Response to authors**
> >
> > Thank you for the explanation.
> > In my opinion, the discussion you provided on dynamic clustering prior works and other techniques for dimension reduction should be added to the paper.
> > Following the author responses, I will maintain my original score.

---

### Official Review · Reviewer_xc3D · 2022-07-10

**Rating:** 5
**Confidence:** 3
**Soundness:** 3 good
**Presentation:** 3 good
**Contribution:** 2 fair

**Summary:**

This work focuses on the euclidean $(k, \ell)$-clustering problem with low average sensitivity. $k$-median and $k$-means are the two of the most studied problems and which are special cases of this work. They focus on the changes to the output distribution in the case that one of the input points is deleted.

They first show that $D^\ell$-sampling has average sensitivity of $O(k/n)$, which is also tight up to a constant factor. They also show a  general reduction from any algorithm to one that has low sensitivity by losing a factor $1+\epsilon$. They also show that their results can be extended to Consistent and Dynamic settings, two very interesting settings in the literature.


After Rebuttal:

I do not find the responses of the authors to the second and third questions that I asked convincing.

2- Basically the authors are arguing that their results is only interesting in a special setting that $\Delta$ is huge and if $k >> log(\Delta)$ their result is worst which is different from the claim in the paper.
3- The argument here is hiding some of the most important factor and focusing only on log with out explaining the reason behind it.

From my point of view, the two main applications of this work are significantly weaker than what presented in the paper. Reading the rebuttal and the other reviews, I updated my score.

**Questions:**

1- Can the hardness result be extended to general $\ell$? Is it possible to argue that the terms in the general reduction are tight?

2- Why is the consistent result interesting given the previous work?

3- Is there something that I am missing regarding the previous works in the Dynamic setting?

4- What are the new tools / ideas in this work? I think that the result is nice even if it is not possible to point out a concrete new technique.


**Limitations:**

I think it is covered, notice that the goal of this paper is not private algorithms.

**Strengths And Weaknesses:**

The paper is in good shape and the results are clear. I found the reduction interesting, it is based on coreset construction. From the analysis it seems that the approach cannot be extended to achieve better results as well. It is always interesting to understand $D^\ell$ sampling better since it is still one of the most basic algorithms for $k$-means. The ideas and techniques are nice but I do not see any major new ones in this work.

I am concerned about the consistent and dynamic settings, it seems that the authors have missed some of the most recent results. For example for consistent clustering, “Consistent k-Clustering for General Metrics” achieves better results than the one cited and also the results in this work. Moreover there are also results in a dynamic setting like “Fully-Dynamic Coresets” which is not considered in this work. From my point of view, the extensions to consistent and dynamic setting needs more work and is not clear and interesting currently.

Typo line 26: even k = 2

---

> ### Author Response · Authors · 2022-08-02
> **Re: Official Review of Paper11263 by Reviewer xc3D**
>
> Thank you very much for the careful reading.
>
>
> 1. The hardness result (Theorem A.1) easily extends to any $\ell$ because we only used the property of the hardness instance that, after deleting one of $x_1,\ldots,x_k$, the optimal solution is unique and its cost is zero. Showing tightness (or improving) the bound of the general reduction is a natural open problem.
> 2. Thanks for sharing the important previous work (Consistent k-Clustering for General Metrics). The bound given there is $O(k \cdot \mathrm{polylog}(n\Delta))$, where $\Delta$ is the aspect ratio of the instance, that is, the ratio of the maximum distance to the minimum distance between pairs of points in the instance. Our bound is $O(k^2 \log n)$ and is incomparable to the previous one. We note that, if $N \geq \max\{\log k, n\}$ is the number of bits used to represent the instance, then the aspect ratio $\Delta$ can be $2^N$. In this case, the previous bound becomes $O(k \cdot \mathrm{polylog}(2^N n)) = O(\mathrm{poly}(N))$, which is huge. Also, another notable difference is that the previous work studies metric $k$-means, and hence output points should be selected from the input points, whereas we study “Euclidean” $k$-means, and hence the output points can be arbitrarily chosen from $\mathbb{R}^d$.
> 3. Thanks for sharing the important previous work (Fully-Dynamic Coresets). Their algorithm maintains a coreset of size $O(\epsilon^{-2}k \log n \log^2 k)$. Hence if we apply a PTAS on the coreset to get a $(1+\epsilon)$-approximate solution, the update time becomes $\mathrm{poly}(n)$ (ignoring dependencies on $\epsilon,k,\ell$). By contrast, our update time is $O(\log^2 n)$ (ignoring dependencies on $\epsilon,k,\ell$). Some other dynamic algorithms are known [1, 2], but their bounds depend on the aspect ratio. Also, another notable difference is that the previous work studies metric $k$-means, and hence output points should be selected from the input points, whereas we study “Euclidean” $k$-means, and hence the output points can be arbitrarily chosen from $\mathbb{R}^d$.
> 4. The idea of using coresets is novel in the design of algorithms with low average sensitivity. Also, the argument used in the transformation from algorithms with low average sensitivity to consistent/dynamic algorithms (such as the use of lazy sampling) is novel.
>
> We will revise the manuscript and incorporate the discussion above.
>
> [1] Bateni, MohammadHossein, et al. "Optimal Fully Dynamic $ k $-Centers Clustering." arXiv preprint arXiv:2112.07050 (2021).
> [2] Goranci, Gramoz, et al. "Fully Dynamic k-Center Clustering in Low Dimensional Metrics." 2021 Proceedings of the Workshop on Algorithm Engineering and Experiments (ALENEX). 2021.

---

### Official Review · Reviewer_94W3 · 2022-07-10

**Rating:** 5
**Confidence:** 4
**Soundness:** 3 good
**Presentation:** 3 good
**Contribution:** 3 good

**Summary:**

This paper considers the average sensitivity of the Euclidean $(k, \ell)$-clustering problem, which measures the stability of the output in total
variation distance against deleting a random point from the input data. The authors first show that D^l-sampling has low average sensitivity, and then show that any approximation algorithm for Euclidean $(k, \ell)$-clustering can be transformed into an algorithm with a low average
sensitivity while almost preserving the approximation guarantee, via a coreset construction approach. They also extend their result to consistent and dynamic settings.

**Questions:**

- Thm 3.1. The authors argue that the algorithm can output a $(1+\varepsilon)\alpha$-approximation. However, it seems that they first do a dimension reduction and then construct a coreset on the reduction space. Then how can we obtain a constant approximation of the original space? A solution to the reduction space does not seem to be what we want.

---------------------------------------------------------------------------------------------------------------------------------------------------------------------------------------
Thanks for the response. It makes sense to me. I have the following additional question accordingly.

The average sensitivity contains an unexpected factor $d$, which seems to come from the coreset size in Theorem 3.4. However, there are several approaches to remove this dependence, e.g.,
- Lingxiao Huang, Nisheeth K. Vishnoi: Coresets for clustering in Euclidean spaces: importance sampling is nearly optimal. STOC 2020: 1416-1429
- Vincent Cohen-Addad, David Saulpic, Chris Schwiegelshohn: A new coreset framework for clustering. STOC 2021: 169-182

A difference is that they apply two-staged importance sampling or partition datasets into rings, which makes it not easy to use the argument of Lemma B.1. However, since you consider coreset construction for 1-clustering, I conjecture that these approaches can help you remove the dimension dependence in the average sensitivity.

**Limitations:**

Not.

**Strengths And Weaknesses:**

Strengths:
Generally, I think the topic is essential and the notion of average sensitivity on Euclidean $k$-clustering is interesting. This paper first provides an analysis of the average sensitivity of Euclidean $k$-clustering, which provides evidence of the stability of the well-studied k-means++ algorithm. The writing is good and explains the contributions well.

Weaknesses:
I have some concerns about the main result Theorem 3.1. See Questions. I may raise my score if the question is answered.

---

> ### Author Response · Authors · 2022-08-02
> **Re: Official Review of Paper11263 by Reviewer 94W3**
>
> Thank you very much for the careful reading. Your question about Theorem 3.1 is very important. Thanks for pointing it out.
>
> [Solution recovery] We can recover a good solution $Z=\{z_1,\ldots,z_k\}$ to the original instance $X$ as follows: Let $Z'=\{z_1',\ldots,z_k'\}$ be the $\alpha$-approximate solution to the instance $X'$ obtained from $X$ by applying the dimension reduction (Lemma 3.2). Then, we compute a partition of $\mathcal{P}=(P_1,\ldots,P_k)$ of $X$ using $z_1',\ldots,z_k'$, that is, we add $x \in X$ to $P_j$ if $z_j'$ is the closest among them. Then for each $j \in \{1,\ldots,k\}$, compute a coreset $C_j$ of $P_j$ and then let $z_j$ be the point that mimizes $\mathrm{cost}_{\{z_j\}}^\ell(C_j)$, that is, we solve Euclidean $(1,\ell)$-clustering on $C_j$. Finally, we output the set $Z=\{z_1,\ldots,z_k\}$, which turns out to be an $O(\alpha)$-approximate solution to the original instance, via the analysis that follows.
>
> [Cost] For a partition $\mathcal{Q}=(Q_1,\ldots,Q_k)$, let $\mathrm{cost}\_{\mathcal{Q}}^\ell(X) = \sum_{j=1}^k \min_v \sum_{x \in Q_j}\|x - v\|^\ell$ be the cost of the partition $\mathcal{Q}$. Let us first show that the partition $\mathcal{P}$ constructed above is a good partition. The dimension reduction used in our algorithm (Lemma 3.2) preserves not only the optimal value but also the cost of any partition: $\mathrm{cost}\_{\mathcal{P}}^\ell(X) \approx \mathrm{cost}\_{\mathcal{P}}^\ell(X')$ for any partition $\mathcal{P}$ (We can view $\mathcal{P}$ as a partition of $X'$ as well, so $\mathrm{cost}\_{\mathcal{P}}^\ell(X')$ is well defined). Then, we have $\mathrm{cost}\_{\mathcal{P}}^\ell(X) \approx \mathrm{cost}\_{\mathcal{P}}^\ell(X') \leq \mathrm{cost}\_{Z'}^\ell(X') \approx \alpha \cdot \mathrm{opt}^\ell(X') \approx  \alpha \cdot \mathrm{opt}^\ell(X)$, as desired. It follows that $\mathrm{cost}\_Z^\ell(X) = \sum_{j=1}^k \mathrm{cost}\_{\{z_j\}}^\ell(C_j) \approx \mathrm{cost}\_{\mathcal{P}}^\ell(X) \approx \alpha \cdot \mathrm{opt}^\ell(X)$.
>
> [Average Sensitivity] Because the average sensitivity of $Z'$ is small as shown by Theorem 3.1, that of the partition $\mathcal{P}$ is also small if we ignore the deleted point. More specifically, let $\mathcal{P} = (P_1,\ldots,P_k)$ and $\mathcal{P}^{(i)} = (P_1^{(i)},\ldots,P_k^{(i)})$ be the output partitions for $X$ and  $X^{(i)}$, respectively. Then, $\sum_{i=1}^n d_{\mathrm{TV}}(\mathcal{P}\setminus \{x_i\}, \mathcal{P}^{(i)})$ is small, where $\mathcal{P}\setminus \{x_i\}=(P_1 \setminus \{x_i\},\ldots,P_k\setminus \{x_i\})$ is the partition obtained from $\mathcal{P}$ by deleting $x_i$. So, let’s assume $\mathcal{P}\setminus \{x_i\} = \mathcal{P}^{(i)}$ happened. Then by Theorem 3.4 and Lemma 3.5, the construction of $C_1,\ldots,C_k$ has average sensitivity  $\tilde{O}(\frac{2^{2\ell}k}{\epsilon^2 n}(d +\log \frac{1}{\delta}))$, and $Z$ has the same average sensitivity. We note that we have $d$ instead of $dk$ in the bound because only one of $C_1,\ldots,C_k$ changes upon deleting $x_i$. So the average sensitivity of Theorem 3.1 should have been $\tilde{O}(\frac{2^{2\ell}k}{\epsilon^3 n}(\frac{\ell^5 k}{\epsilon^2} + d +\log \frac{1}{\delta}))$. If we do not apply dimension reduction the bound will be $O(dk^2/n)$ (ignoring $\ell, \epsilon, \delta$) (which might be also fine).
>
> We will add the discussion above to the manuscript.

---

### Meta-Review · Area_Chair_sarW · 2022-08-30

**Recommendation:** Accept
**Confidence:** Certain

**Metareview:**

I agree with the reviewers that the topic is essential, the notion of average sensitivity on Euclidean -clustering is interesting, and that the paper addresses important problem that is very relevant for the inevitability noisy data in the big data era.
As complained, the "dynamic data" section is a bit misleading compared to previous work and I suggest to remove it.
Please also add the discussion in the rebuttal to the paper or at least the supp. material.

**Award:**

No

---

### Decision · Program_Chairs · 2022-09-14

Accept